# Psychometric Validation of the Greek Version of the Scale “Temper Loss” of the Questionnaire “Multidimensional Assessment Profile of Disruptive Behavior (MAP-DB)”

**DOI:** 10.3390/children9091328

**Published:** 2022-08-31

**Authors:** Maria Micha, Gerasimos Makris, Maria Michou, Christina Kanaka-Gantenbein, Lauren Wakschlag, Panagiota Pervanidou

**Affiliations:** 1Postgraduate Program “The Science of Stress and Health Promotion”, School of Medicine, National and Kapodistrian University of Athens, 11527 Athens, Greece; 2Unit of Developmental and Behavioral Pediatrics, First Department of Pediatrics, School of Medicine, National and Kapodistrian University of Athens, “Aghia Sophia” Children’s Hospital, 11527 Athens, Greece; 3Human Ecology Laboratory, Department of Home Economics and Ecology, Harokopio University, 17676 Kallithea, Greece; 4First Department of Pediatrics, School of Medicine, National and Kapodistrian University of Athens, “Aghia Sophia” Children’s Hospital, 11527 Athens, Greece; 5Department of Medical Social Sciences, Feinberg School of Medicine and Institute for Innovations in Developmental Sciences, Northwestern University, Chicago, IL 60611, USA

**Keywords:** irritability, temper tantrums, disruptive behavior, developmental psychopathology

## Abstract

It has become increasingly evident that vulnerability to psychopathology is identifiable early in life. A body of evidence suggests that the recognition and prevention of a spectrum of typical/atypical behaviors in preschoolers can lead to earlier diagnosis and treatment. The Multidimensional Assessment Profile of Disruptive Behavior (MAP-DB) is a parent-completed instrument that has been developed recently to differentiate normative misbehaviors in early childhood from markers of clinical risk. The aim of the present study was to validate the “Temper Loss” Subscale in the Greek language and to assess its psychometric properties in healthy children. An on-print parent-report survey was conducted among Greek children, aged between 3 and 5.5 years. The survey included the MAP-DB and the Child Behavior Checklist (CBCL). A total of 400 respondents participated in the study. The analysis suggested that the Greek version of the MAP-DB’s “Temper Loss” scale has good psychometric properties. The results of the Exploratory Factor Analysis (EFA) of the 22 items were able to explain 64.4% of the total variance. Internal consistency for the one subscale was satisfactory, with Cronbach’s alpha at 0.970. This scale can be used by researchers and practitioners for the evaluation of dimensional phenotypes in early childhood.

## 1. Introduction

The term “irritability” has been used to describe a mood with substantial stability across time that manifests as a reaction to negative emotional stimuli, including frustration [1]. It is a criterion for the diagnosis of many disorders included in the Diagnostic and Statistical Manual of Mental Disorders of the American Psychiatric Association (DSM-5), such as Anxiety Disorders, Mood Disorders, and Disruptive Behavior Disorder [2]. Of note, the main symptom of Disruptive Mood Dysregulation Disorder (DMDD) is chronic irritability [3]. In addition to its transdiagnostic utility for predicting both internalizing and externalizing psychopathological disorders, measures of irritability can be used to predict suicide, academic problems, and income in adulthood [4,5,6].

Irritability has both behavioral and emotional components in irritable moods and tantrums [7]. It is also differentiated from both kinds of anger, which characterizes an emotional state and aggression, which is a type of behavior [7]. The concept of anger is used to describe a situation which is due to the activation of the autonomic nervous system, is characterized by emotions that manifest in different ways and intensity and has an impact on both psychological and biological levels [8,9]. Manifestation of anger in the context of irritability often occurs in response to frustration [10], while it is also associated with striving to achieve difficult goals [10,11,12].

Regarding the component of aggression (i.e., a verbal or motor behavior that arises from anger or frustration) [13,14] that is intended to cause harm to another person [15], it can be categorized into reactive and preventive aggression depending on the type of the stimulus that caused its manifestation [16,17]. However, not all irritants will show aggression. In fact, a recent review suggested that 12% of children with outbursts of anger exhibit aggressive behavior in the general population [18]. It is worth noting, that in order for irritability to be considered a pathological dimension, it must be at a level that is not consistent with the developmental level of the child. In young children, irritability, particularly tantrums, is a common behavioral expression of normative misbehavior. Wakschlag et al., 2010 reported that more than 80% of preschool children had temper tantrums in the last month [19]. However, features of early irritability that predict impairing psychopathology have been identified with the “Multidimensional Assessment Profile of Disruptive Behavior (MAP-DB)” questionnaire, including dysregulation, and developmentally unexpectable and contextual factors. Elevated irritability in young children has transdiagnostic utility for predicting internalizing and externalizing problems as well as neural alterations [20,21,22].

Developmentally, irritability usually decreases with age, and from adolescence onwards, high levels of irritability become less pronounced and more predictive of difficulties [23]. The developmental difference in regulatory levels of irritability should be taken into account when defining clinical irritability in children and adolescents.

The manifestation of irritability is a common phenomenon among children and adolescents and affects 3% of the general population [9]. However, in another study evaluating the intensity, frequency, and duration of irritability, an unusually high incidence of 20% was recorded in adolescents [24]. It is a fact that irritable young people, compared to healthy people, tend to interpret vague environmental stimuli as threatening, thus increasing the likelihood of expressing feelings of anger and aggression [4].

A wide range of psychopathological disorders, including disruptive behavior, attention deficit hyperactivity disorder (ADHD), depression, and anxiety can be predicted by assessing the manifestation of preschool irritability [6,25]. At the same time, a correlation has been shown between the early onset of irritability and the late onset of depressive and anxiety disorders in studies involving preschool and school-aged children [26,27,28].

It is therefore important to identify clinically salient irritability in young children, thus leading to earlier identification and preventative diagnoses and treatment. In fact, the MAP-DB questionnaire provides an innovative method for defining developmental dimensional phenotypes in early childhood [29]. It was designed to distinguish the normal manifestation of misbehavior during preschool age from the clinically significant risk markers by using an objective frequency rating scale, including both normatively occurring and rare, more severe expressions and assesses behavior across varying contexts that differ in the extent to which the behavior is developmentally expected. In order to determine the spectrum of irritability in preschool children, the scale of “Temper Loss”, which consists of 22 questions, was used. This scale can be used to successfully predict the occurrence of anxiety disorders, mood disorders, and disruptive mood dysregulation disorder [20], while also being distinguished for its high reliability and remarkable psychometric properties [7,30]. The purpose of this study was to validate the Greek version of the scale “Temper Loss” of “Multidimensional Assessment Profile of Disruptive Behavior (MAP-DB)” questionnaire in preschool children by determining its unidimensional structure, reliability, and criterion validity.

## 2. Materials and Methods

### 2.1. Translation Procedure

Permission for translation was requested by Lauren S. Wakschlag who developed the MAP-DB questionnaire. The translation process was undertaken by an experienced panel following the World Health Organization principles [31]. After numerous reflective steps conducted by the panel, confusing sentences were located with the execution of a test-pretest of the research tool. The sample consisted of parents (15 men, 25 women) of preschool children attending a nursery school, as this choice seemed to be representative of the population of the study regarding mother language and age span. After these procedures, the finalization of the Greek version of the questionnaire was performed. All the rights of the MAP-DB questionnaire belong to the authors.

### 2.2. Participants and Procedures

The study was conducted in the area of Attica, Greece from October 2021 to February 2022. A total of 400 children (239 boys and 161 girls), aged between 3 and 5.5 years, were included in the study. More specifically, in order to recruit volunteers, the first basic information tank was the individual municipalities, which maintain official registration documents of the population attending the preschools as well as the kindergartens that belong to them. Relevant letters were sent informing the bodies about the conduct of this investigation and the need for their assistance. Children with neurodevelopmental and/or intellectual disabilities and/or other psychiatric diseases, genetic syndromes, chromosomal abnormalities, severe prematurity, metabolic disorders, and neurological or other chronic diseases were excluded from the study. Of the 480 individuals that were approached for screening, 426 (88.7%) completed the entire questionnaire. Of these 426 individuals, 400 children were eligible. The main reasons for ineligibility were poor use of the Greek language (N = 2) and reported use of services for developmental delays (N = 24). The nationality of the participants was not an exclusion criterion, since parents spoke fluently the Greek language fluently. The descriptive characteristics of the participants are presented in Table 1.

### 2.3. Ethical Considerations

The study’s protocol was approved by the ethics committee of the Medical School of the National and Kapodistrian University of Athens and was in accordance with the Declaration of Helsinki (2013). For the completion of the printed questionnaire, volunteers who participated in the study, signed a form of informed consent and confidentiality.

### 2.4. Measures

Participants were asked about their sex, age, national identity, family and employment status, educational level, as well as the sex and age of their child.

Multidimensional Assessment Profile of Disruptive Behavior (MAP-DB)–Temper Loss: MAP-DB/TL provides an innovative method for distinguishing the normal-abnormal spectrum of phenotypes in early childhood [29]. It was designed to examine normative misbehaviors from atypical behaviors in terms of normative expressions, such as tantrums in the face of frustration, versus dysregulated tantrums in the view of regularity and severity [20]. It is divided into 4 sections (temper loss, noncompliance, aggression, and low concern for others) and includes 6 possible answers [0 = never, 1 = rare (less than once a week), 2 = some (1–3) days of the week, 3 = most (4–6)] days of the week, 4 = daily, 5 = many times in the same day].

The Child Behavior Checklist for Ages 1.5–5 (CBCL/1½–5): This parent-completed questionnaire is used to measure the functionality, sociability, and the existence of psychopathology in children. The CBCL/1½–5 is a component in the Achenbach System of Empirically Based Assessment (ASEBA). It consists of 100 items, 99 of which are closed, and one of which is open-ended adding additional issues not listed above. Based on the American standardization, the lower level entails 100 items with scores ranging from 0 to 2 on emotional, behavioral, and social competence [32]. The following level consists of psychiatric syndrome scales based on confirmatory and exploratory factor analysis. On the next level, 36-item “internalizing” and 24-item “externalizing” scales are included. Internalizing and externalizing problems are not mutually exclusive. The final level consists of the Total Problem Scale (TPS) [32] whose score is calculated by adding the total of 0-1-2- scores on the 99 specific problem items and the highest value of 1 or 2 on any problems entered for item 100. Therefore, the TPS has a range of 0 to 200. In addition, in order to relate symptomatology to diagnostic criteria, taking into consideration cross-cultural comparisons, five different diagnostic tools were constructed by Achenbach & Rescorla [32]. These were Affective Problems, Anxiety Problems, Pervasive Developmental Problems, Attention Deficit/Hyperactivity Problems, and Oppositional Defiant Problems. The Greek version of the CBCL [33] was used.

## 3. Results

Data are presented as N (%) for qualitative variables and as mean (SD) for quantitative variables. Exploratory Factor Analysis (EFA) was conducted for the 22 items. Cronbach’s alpha was calculated to examine internal consistency. Independent samples *t*-test and ANOVA test were conducted to evaluate differences between groups. Pearson’s rho coefficient was used to assess correlations between quantitative variables. SPSS v.24 for Windows was used to perform statistical analyses and the level of significance for all analyses was 0.05.

A total of 400 valid responses were collected. Participants’ sociodemographic characteristics and descriptive statistics for the MAP -DB “Temper Loss” subscale and the CBCL/1½–5 scale scores are presented in Table 1. In total, the median age of the respondents was 36 years while the majority of children were females, and their median age was 4 years. The majority of participants were females (84.0%), had a Bachelor’s degree (74.8%), married (78.8%), Greek (82.2%), and employed (80.0%) (Table 1).

Table 2 presents descriptive statistics for the MAP-DB’s “Temper Loss” scale and seven subscales of the CBCL/1½–5.

Kaiser-Meyer-Olkin (KMO) test and Barlett’s Sphericity Test were used in order to assess the adequacy of the sample. KMO value was 0.947 and the significance of Bartlett’s test of sphericity was *p* < 0.001. EFA of the 22 items with direct oblique rotation is presented in Table 3. The Scree plot, as a result of the EFA, indicated that one factor exceeded the Eigenvalue 1 (Figure 1). It was observed that the 22 items were able to explain 64.4% of the total variance. The one factor that was examined was named as temper of loss. Cronbach’s α coefficient for this was 0.970, indicating satisfactory internal consistency. All item–total correlations were >0.580.

Correlations between Temper loss and the seven subscales of the CBCL checklist are presented in Table 4. Overall, a strong positive correlation between the subscale of “temper loss” and emotionally reactive and aggressive behaviors manifesting that irritability is associated with externalizing problems. The Spearman’s correlation coefficient between the Temper loss, emotionally reactive, and aggression subscale was 0.44, 0.40, and 0.80, respectively. Overall, a strong positive correlation between the three variables manifests that irritability is associated with externalizing problems. The Spearman’s correlation coefficient between emotionally reactive and anxious-depressed, somatic complaints and emotionally reactive, and somatic complaints and anxious-depressed subscales were 0.78, 0.69, and 0.68, respectively. Correlations are also made between the two general subscales (internalizing/externalizing scale). Specifically, the Spearman’s correlation coefficient between attention problems and emotionally reactive, attention problems and anxious-depressed, attention problems and somatic complaints, aggressive behavior and emotionally reactive, and finally aggressive behavior and anxious-depressed were 0.67, 0.68, 0.83, 0.84, 0.76, respectively.

Table 5 presents associations between temper loss and the study variables. Statistically significant correlations were found between sex of parents and the subscale of MAP-DB (*p* < 0.0001). From the MAP-DB scale, temper loss was positively associated to parents’ sex (*p* < 0.0001), and to parents’ nationality (*p* < 0.0001). Also, the temper of loss score was positively associated to sex of children (*p* = 0.001).

## 4. Discussion

This study aimed to adapt and validate the parent-reported subscale “Temper loss” of the MAP-DB questionnaire in Greek language and population. The development of the MAP-DB questionnaire addressed the need of measuring a child’s normative misbehavior from abnormal behaviors within a developmental context [7]. Specifically, the subscale of temper loss verifies the need to separate irritability from anger and aggression, because irritability can exist even without the existence of aggressive acts. On the other hand, irritability may be an elemental predictive indicator of psychopathology [1,30]. The present study revealed significant reliability in the validation of the psychometric properties of the MAP-DB’s Temper loss scale. This scale was developed following a thorough process, that combined theoretical delineation of its dimensions, multiple validation samples, focus groups, and item selection informed by Item Response Theory (IRT). As such, it has strong dimensionality and sensitivity [34]. The dimension provided good internal consistency (a = 0.97). This finding is in line with the original results of Lauren S. Wakschlag [7,30].

Studies of irritability in children and adolescents appear to have increased significantly in the last decade [4]. The manifestation of irritability is a common phenomenon among children and adolescents and affects 3% of the general population [9]. The concept of irritability was introduced in 1957 by Buss & Durkee (1957) in the “Buss-Durkee Hostility Inventory” which included an overall assessment of aggression. Based on the responses of 1000 participants, an irritability questionnaire was created and irritability was established as a feature [35]. Various irritability scales were then published to assess several parameters around the manifestation and management of anger and reactive aggression [2,34].

The expression of irritability through episodes of anger and reactive aggression is often observed from infancy to childhood [36,37]. Specifically, Lewis et al., 1990 reported that facial expressions of sadness or anger due to frustration are already evident from the age of two months and differ between infants [11]. It is worth noting that the stability of expression and regulation of reactive aggression and anger manifested in infancy is limited and is not able to successfully predict the onset of behavioral problems in preschool children [11].

Developmentally, outbursts of anger peak in preschool age [30]. Gaining self-control [38] and increasing social skills [39] in most cases lead to a reduction in aggression during the transition from preschool to school age [36,40,41]. The innate tendency to manifest irritability is greatly influenced by environmental factors and in particular, those related to the family environment, such as lack of response to the child’s grief, maternal depression, and upbringing in a hostile environment [40,42,43].

Regarding the causes of irritability, the effect of genetic and environmental factors on the manifestation of irritability seems to be similar, as the heritability in adolescents and adults has been estimated at around 30–40% [44,45,46]. Furthermore, it has been shown that the genetic influence on irritability is not constant. Differences are even observed between the two sexes, with girls showing reduced heritability from early childhood to adulthood, while boys show the opposite [47].

Scales measuring irritability are one of the most appropriate ways of assessing it, which should be formulated in such a way as to enable the recognition of a clinically significant type of irritability, thus leading to better diagnosis and treatment [34]. The Affective Reactivity Index (ARI) provides concisely an innovative method for defining irritable mood rather than its behavioral consequences [48]. It was designed to examine irritability over the last 6 months, which means that the focus of the scale is on chronic irritability [32]. It is divided into three sections: (a) threshold for an angry reaction; (b) frequency of angry feelings/ behaviors; (c) duration of such feelings/behaviors. This scale can be used to successfully predict the occurrence of irritable mood, while also being distinguished for its high reliability and remarkable psychometric properties [48]. On the other hand, the MAB-DB questionnaire as a whole was not a specifically examined measure for irritability; however, the MAP-DB’s subscale of Temper loss contains 22 items that assess normative misbehaviors, in terms of tantrums, and dysregulation in preschool children [20].

To add, positive strong correlations were found between the “temper loss” scale of the MAP-DB and the CBCL, regarding the assessment of the preschool child’s behavior. Although we do not know how predictive of the manifestation of psychopathology later in the life is the data of the participants, it should be noted that it is vital to identify a child’s behavioral, emotional, and social problems and to offer timely and valid help to both the children and the family in order to anticipate any manifestations of secondary problems. Furthermore, it should be noted that severe behaviors are reported either as internal or external problems, which means that they do not overlap in these questionnaires [32,33].

The present study has several limitations. To begin with, our results should be interpreted with caution, since the current sample is not representative of the general population. Additionally, irritability was assessed using a single questionnaire, and no other specific tool was used. Finally, the number of participants did not allow us to validate the entire questionnaire.

However, the study of irritability in preschool children remains important, as it is possible to determine the developmental spectrum and the evolution of chronic irritability from childhood through adolescence and adulthood. In addition, the identification of the relations between irritability and a child’s psychopathological traits can contribute to prevention and early intervention. Future research may focus on the validation of the entire MAP-DB questionnaire in healthy children, as well as in children with signs of neurodevelopmental disorders.

Indeed, children with autism spectrum disorder (ASD) are characterized by high levels of irritability and aggressive behavior, which greatly affect their lives and their families. A study by O’Donnell et al., 2012 found that preschool children (3–4 years old) with ASD face difficulties in processing sensory stimuli, and these difficulties are directly related to the manifestation of behavioral problems, including irritability [49]. In addition, irritability is likely one of the factors linking ADHD to the later onset of depressive disorder in the general population, while children with ADHD and coexisting persistent irritability have been found to be at high risk of developing depressive symptoms [50]. Also, a study by Mikita et al., 2015 found that 10 to 16-year-old boys with high-functioning ASD exhibited greater levels of irritability than typically developing boys and showed marked changes in cortisol levels and heart rate in response to stressful stimuli [51]. This is because high levels of stress over a long period of time have been associated with high levels of cortisol in the body [52]. Therefore, in the future, the correlation of irritability with behavioral parameters and neurobiological indicators of stress in preschool children with irritability and features of neurodevelopmental disorders could be investigated and shown, as well as the differences in irritability and related neurological indicators in children with neurodevelopmental disorders compared to typically developing children. 

In conclusion, MAP-DB is a newly introduced instrument aiming to evaluate operationalizing developmentally specified, dimensional phenotypes in early childhood [20,29]. This is the first study validating the Temper Loss scale of the MAP-DB in preschool children in Greece, which can be used by researchers and practitioners. Τhe results of the present study provide evidence for the validity of using the Greek version of the MAP-DB’s Temper Loss scale in preschool children. 

## Figures and Tables

**Figure 1 children-09-01328-f001:**
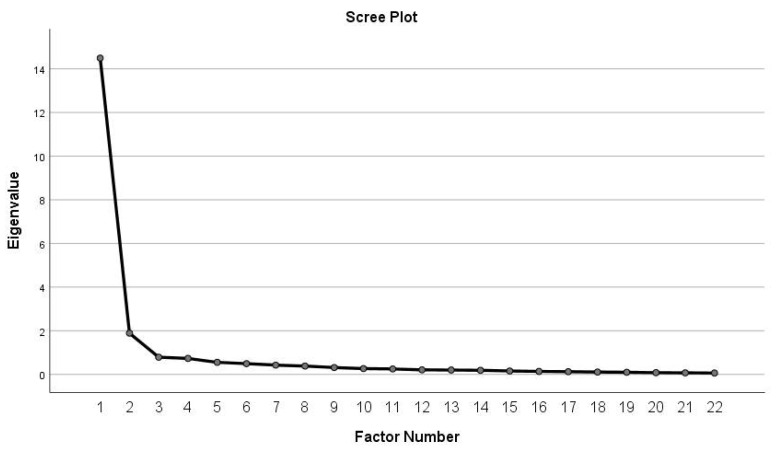
Depicts the Scree plot, as a result of the EFA.

**Table 1 children-09-01328-t001:** Samples’ Sociodemographic characteristics (N = 400).

**Sex of Children N (%)**	
Males	239 (59.8)
Females	161 (40.3)
**Sex of Parents N (%)**	
Males	64 (16.0)
Females	336 (84.0)
**Age of Children (years)**	
Median (IQR)	4.0 (1.5)
Mean (SD)	4.1 (0.8)
**Age of Parents (years)**	
Median (IQR)	36.0 (9.0)
Mean (SD)	36.6 (5.4)
**Parents’ Educational Status N (%)**	
Until Upper Secondary School (Lyceum)	77 (19.3)
Bachelor	299 (74.8)
MSc/PhD	24 (6.0)
**Parents’ Marital Status N (%)**	
Married	315 (78.8)
Separated	45 (11.3)
Divorced	37 (9.3)
Death of one or both parents	3 (0.8)
**Parents’ Nationality N (%)**	
Greek	328 (82.0)
Albanian	50 (12.5)
Bulgarian	17 (4.3)
Other	5 (1.3)
**Parents’ Work Status N (%)**	
Employed	320 (80.0)
Unemployed	51 (12.8)
Part time job	27 (6.8)
Other	2 (0.5)

**Table 2 children-09-01328-t002:** Descriptive statistics for the MAP-DB’s “Temper Loss” scale and seven subscales of the CBCL/1½–5 (N = 400).

Subscales Scores	Mean (SD)
CBCL–Emotional Reactive	60.7 (9.4)
CBCL–Anxious/Depressed	53.1 (4.5)
CBCL–Somatic Complaints	54.5 (5.4)
CBCL-Withdrawn	51.8 (3.5)
CBCL–Sleep Problems	50.3 (1.1)
CBCL–Attention Problems	55.8 (7.5)
CBCL–Aggression Behavior	58.4 (10.5)
MAP-DB A (Temper of loss)	59.5 (32.1)

**Table 3 children-09-01328-t003:** Rotated factor loadings of the Exploratory Factor Analysis (EFA) for the 22 items of the Subscale ‘temper of loss’ of the MAP-DB questionnaire (N = 400) item-total correlation and Cronbach’s alpha coefficients.

	Temper Loss	Item-Total Correlations
10. Have difficulty calming down when angry	0.864	0.835
14. Become frustrated easily	0.786	0.747
17. Get extremely angry	0.832	0.829
25. Yell angrily at someone	0.554	0.580
31. Stamp feet or hold breath during a temper tantrum	0.769	0.770
35. Lose temper or have a tantrum when frustrated, angry, or upset	0.805	0.771
36. Lose temper or have a tantrum when tired, hungry, or sick	0.770	0.727
42. Lose temper or have a tantrum to get something s/he wanted	0.872	0.836
44. Act irritable	0.770	0.755
46. Lose temper or have a tantrum during daily routines	0.771	0.748
49. Break or destroy things during a temper tantrum	0.597	0.627
51. Have a temper tantrum, fall-out, or melt-down	0.938	0.925
57. Stay angry for a long time	0.855	0.827
58. Have a temper tantrum lasting >5 min.	0.922	0.909
63. Lose temper or have a tantrum with other adults	0.773	0.734
64. Lose temper or have a tantrum with parents	0.852	0.834
67. Keep on having a temper tantrum even when you tried to help calm down	0.849	0.837
72. Have a hot or explosive temper	0.842	0.820
74. Have a temper tantrum until exhausted	0.835	0.839
75. Have a short fuse	0.848	0.837
77. Lose temper or have a tantrum “out of the blue”	0.842	0.844
81. Hit, bite, or kick during a temper tantrum	0.577	0.599
% of Variance	64.405	-
Eigenvalues	14.169	-
Alpha of scale	0.970	-

**Table 4 children-09-01328-t004:** Correlations (Spearman’s rho) between CBCL subscales and scale of MAB-DB (N = 400).

	Parent’s Age	Child’s Age	Emotional Reactive	Anxious/ Depressed	Somatic Complaints	Withdrawn	Sleep Problems	Attention Problems	Aggressive Behavior	Temper Loss
Parent’s age	1.000									
Child’s age	0.233 **	1.000								
Emotional Reactive	0.100 **	−0.092	1.000							
Anxious/ Depressed	0.074	−0.108 *	0.785 **	1.000						
Somatic Complaints	0.035	−0.133 **	0.696 **	0.685 **	1.000					
Withdrawn	−0.019	−0.097	0.469 **	0.441 **	0.469 **	1.000				
Sleep problems	0.013	−0.051	0.365 **	0.403 **	0.468 **	0.265 **	1.000			
Attention problems	0.017	−0.105 *	0.674 **	0.686 **	0.830 **	0.390 **	0.361 **	1.000		
Aggressive behavior	0.073	−0.100 *	0.842 **	0.769 **	0.778 **	0.556 **	0.444 **	0.707 **	1.000	
Temper loss	0.080	−0.014	0.441 **	0.376 **	0.293 **	0.258 **	0.171 **	0.299 **	0.406 **	1.000

* correlation is significant at the 0.05 level (2-tailed). ** correlation is significant at the 0.01 level (2-tailed).

**Table 5 children-09-01328-t005:** Differences in sociodemographic characteristics and study’s measurements in the subscale of the MAP-DB questionnaire.

	Categories	Temper of Loss
Sex of children Mean (SD)	Males	50.8 (24.6)
	Females	42.4 (26.3)
	*p*-Value	0.001
Sex of parents Mean (SD)	Males	60.2 (24.8)
	Females	45.0 (25.0)
	*p*-Value	<0.0001
Age of children (years)	Spearman rho	−0.014
	*p*-Value	0.785
Age of parents (years)	Spearman rho	0.080
	*p*-Value	0.115
Parents’ Marital Status Mean (SD)	Married	45.9 (24.9)
	Separated	54.1 (25.9)
	Divorced	49.3 (29.6)
	Death of one or both parents	81.3 (0.6)
	*p*-Value	0.011
Parents’ Educational status Mean (SD)	Until Upper Secondary School (Lyceum)	52.0 (23.1)
	Bachelor	46.3 (25.8)
	MSc/PhD	45.0 (29.7)
	*p*-Value	0.404
Parents’ nationality Mean (SD)	Greek	44.7 (25.9)
	Albanian	59.2 (21.9)
	Bulgarian	60.8 (15.9)
	Other	63.2 (22.6)
	*p*-Value	<0.0001
Parents’ work status Mean (SD)	Employee	47.1 (25.6)
	Unemployed	51.9 (25.8)
	Part time job	43.0 (25.3)
	Other	24.5 (9.1)
	*p*-Value	0.236

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
