# Peer review of "Psychometric Validation of the Greek Version of the Scale “Temper Loss” of the Questionnaire “Multidimensional Assessment Profile of Disruptive Behavior (MAP-DB)”"

_children, 2022, doi:10.3390/children9091328_

Round 1

Reviewer 1 Report

In my opinion, the study is very interesting and worth being published. In fact, the article is well-structured, the method used is congruent with the purpose of the study and the results are presented clearly. 

Nevertheless, the Discussion and Conclusion sections are not sufficiently developed to exhibit the value of the research undertaken by the authors. For this reason, I feel I can give the following suggestions:

- First of all, I suggest to include one short paragraph summarizing the purpose of the study and the main findings obtained.
- Second, I think that the whole discussion needs to be re-structured to include: theoretical implications of the study, practical implications, limitations and future studies and, please, include a brief conclusion (in a separate point or independent section).

I think the authors can easily follow the suggestions I have given in this review and make a new version of their interesting paper.

All best wishes.

Author Response

COMMENTS: In my opinion, the study is very interesting and worth being published. In fact, the article is well-structured, the method used is congruent with the purpose of the study, and the results are presented clearly. 

Nevertheless, the Discussion and Conclusion sections are not sufficiently developed to exhibit the value of the research undertaken by the authors. For this reason, I feel I can give the following suggestions:

RESPONSE: We thank the Reviewer for their comments and suggestions.

Please, find our answers below:

First of all, I suggest to include one short paragraph summarizing the purpose of the study and the main findings obtained.

Reply: Thank you, we have added a paragraph summarizing the findings of the study, in the beginning of the Discussion, highlighted in yellow (lines 220-229)

Second, I think that the whole discussion needs to be re-structured to include: theoretical implications of the study, practical implications, limitations and future studies and, please, include a brief conclusion (in a separate point or independent section).

Reply: We thank the Reviewer, and we have restructured the Discussion accordingly.

Reviewer 2 Report

The theme and contents of this article are interesting and highly relevant for the psychometric validation of the Greek version of the subscale “Temper loss” of the MAP-DB. The article is interesting, well founded and written, and has a careful and detailed description. Despite the article
being a good theoretical and practical contribution to the adaptation, validation of the subscale and
determinate the psychometric properties in healthy children aged between 3 and 5.5 years, 

The theme and contents of this article are interesting and highly relevant for the psychometric validation of the Greek version of the subscale “Temper loss” of the MAP-DB. The article is interesting, well founded and written, and has a careful and detailed description. Despite the article being a good theoretical and practical contribution to the adaptation, validation of the subscale and determinate the psychometric properties in healthy children aged between 3 and 5.5 years, it would be important to review some specific aspects explained below: Abstract: There is a space error (line 24). Keywords: There is an error in the “irritability” (line 33).

1. Introduction • It seems more adequate refer the Likert Scale of the MAP-DB in Method Scale and excluded this content from this section (see lines 93, 94). • Grammatical formulation: one “and” that are dispensable (see line 80). • There are space errors (see lines 61, 66, 84).

2. Materials and Methods 2.2. Participants and procedures • There are space errors (see lines 115, 116, 120). 2.3. Ethical considerations • There is a space/punctuation error (see line 131). 2.4.Measures • There are space errors (see lines 134, 137, 139). • The Likert Scale, with 6 possible answers that are described in Introduction (see 92-94), must be inserted in this section.

3. Results • There is a space error (see line 163). • Include 18.1% participants that are not Greek (psychometric validation of the Greek version of the “temper Loss”). Explain why it was chosen to keep these non-Greek participants. •Convergent and divergent validity (with other standardized tests measuring preschoolers’ developmental dimension of irritability and personality traits) are not explored in this study although the authors identify its relevance in the suggestions for future research. •A table with the items of the MAP-DB/Temper Loss in Greek/English should be included in the article. • Indicated the range of the Eigenvalues of the items of the factorial analyses. • The KMO and Bartlett´s test of sphericity results should be presented, as well as the Scree plot. • It is written that “Principal component analysis (PCA) was conducted to extract the factors of the MAP-DB scale.” (see lines 162-163). Later, it is said that “The results of the Exploratory Factor Analysis (EFA) (…) are presented in table 3 (see lines 179, 180). To design table 3 refer “(…) of the principal components analysis (PCA) (...)” (lines 184). Clarify what analysis was used because two different are reported.

4. Discussion • There are space errors (see line 234, 243, 253, 256). • Paragraph error (see line 225). • Format: “(…) Buss & Durkee (1957) (…)”; “(…) results of Lauren S. Wakschlag… (…)” (see lines 206, 254). It´s not very clear what format is used in this article its important be coherent in the use of format suggested by children - MPDI. • The psychometric validation of the Greek version of the “temper Loss” in clinical population should be mentioned in the suggestions for future research. References • There should be several bibliographic references, on the relevant content, from the last 3 years (i.e., 2022, 2021, 2020). The present article only has two. There is also only one bibliographic reference from 2018 and zero from 2019. The contents of investigations from the last 5 years should be included in the article. • There are references that are incomplete (e.g., don´t have de year of publication or/and number of publication or/and pages) (see lines 319, 333, 381). 

Author Response

REVIEWER 2

We thank the Reviewer for their comments and suggestions! Please, find our comments below:

The theme and contents of this article are interesting and highly relevant for the psychometric validation of the Greek version of the subscale “Temper loss” of the MAP-DB. The article is interesting, well founded and written, and has a careful and detailed description. Despite the article being a good theoretical and practical contribution to the adaptation, validation of the subscale and determinate the psychometric properties in healthy children aged between 3 and 5.5 years, it would be important to review some specific aspects explained below:

Abstract: There is a space error (line 24). Keywords: There is an error in the “irritability” (line 33).

Reply: We have corrected these errors, thank you!

Introduction: • It seems more adequate refer the Likert Scale of the MAP-DB in Method Scale and excluded this content from this section (see lines 93, 94). • Grammatical formulation: one “and” that are dispensable (see line 80). • There are space errors (see lines 61, 66, 84).

Reply: Thank. You, we have corrected all errors. The description of the MAP-DB has moved to the Methods (lines 146-148).

Materials and Methods 2.2. Participants and procedures • There are space errors (see lines 115, 116, 120). 2.3. Ethical considerations • There is a space/punctuation error (see line 131). 2.4.Measures • There are space errors (see lines 134, 137, 139). • The Likert Scale, with 6 possible answers that are described in Introduction (see 92-94), must be inserted in this section.

Reply: Thank you, we have corrected all errors.

Results • There is a space error (see line 163). • Include 18.1% participants that are not Greek (psychometric validation of the Greek version of the “temper Loss”). Explain why it was chosen to keep these non-Greek participants. •Convergent and divergent validity (with other standardized tests measuring preschoolers’ developmental dimension of irritability and personality traits) are not explored in this study although the authors identify its relevance in the suggestions for future research. •A table with the items of the MAP-DB/Temper Loss in Greek/English should be included in the article. • Indicated the range of the Eigenvalues of the items of the factorial analyses. • The KMO and Bartlett´s test of sphericity results should be presented, as well as the Scree plot. • It is written that “Principal component analysis (PCA) was conducted to extract the factors of the MAP-DB scale.” (see lines 162-163). Later, it is said that “The results of the Exploratory Factor Analysis (EFA) (…) are presented in table 3 (see lines 179, 180). To design table 3 refer “(…) of the principal components analysis (PCA) (...)” (lines 184). Clarify what analysis was used because two different are reported.

Reply: We have corrected all errors.

-Regarding the 18% of non-Greek participants, the aim of the study was to evaluate the subscale of ''Temper loss'' of the MAP-DB questionnaire in preschool children in Greece. The nationality of the children and parents was not an exclusion criterion from the research, since parents spoke fluently the Greek language and children were exposed to Greek language.  Indeed, the inclusion of nationalities in the study was based on population distribution, according to published data from the 2001 population census. We have inserted now this information in lines 125-127

- Indeed, we didn’t explore convergent and divergent validity using other measures. This is mentioned in the limitations (lines 281-292)

-The table has been added in the supplementary materials.

- Eigenvalues is presented in Table 3.

-The Reviewer is right, we have now inserted the KMO and test of sphericity (lines 188-193) and the scree plot is presented tin Figure 1.

-It is Exploratory Factor Analysis (EFA), this has been now corrected.

Discussion • There are space errors (see line 234, 243, 253, 256). • Paragraph error (see line 225). • Format: “(…) Buss & Durkee (1957) (…)”; “(…) results of Lauren S. Wakschlag… (…)” (see lines 206, 254). It´s not very clear what format is used in this article its important be coherent in the use of format suggested by children - MPDI. • The psychometric validation of the Greek version of the “temper Loss” in clinical population should be mentioned in the suggestions for future research. References • There should be several bibliographic references, on the relevant content, from the last 3 years (i.e., 2022, 2021, 2020). The present article only has two. There is also only one bibliographic reference from 2018 and zero from 2019. The contents of investigations from the last 5 years should be included in the article. • There are references that are incomplete (e.g., don´t have de year of publication or/and number of publication or/and pages) (see lines 319, 333, 381). 

Reply: we have corrected the errors.

-The validation in clinical population, as a future direction, is now mentioned in the Discussion (lines 299-289) and later, the next paragraph refers to clinical data.

-We have corrected the References (Reference list in yellow) and we have also added additional more recent  citations (Reference list in yellow)

Reviewer 3 Report

Dear Author, 

Nothing to change. Interesting work! Accept! 

Author Response

We thank the Reviewer for their comments.